# Generative News Recommendation

Submission Id: 733*

## ABSTRACT

Most existing news recommendation methods tackle this task by conducting semantic matching between candidate news and user representation produced by past clicked news. However, they ignore the higher-level associative relationships between news, and building these relationships typically requires common-sense knowledge and reasoning ability. And the definition of these methods dictates that they can only deliver news articles as-is. On the contrary, integrating several relevant news into a coherent narrative would assist users in gaining a quicker and more comprehensive understanding of events. In this paper, we propose a novel generative news recommendation paradigm that includes two steps: (1) Leveraging the internal knowledge and reasoning capabilities of the Large Language Model (LLM) to perform high-level matching between candidate news and user representation; (2) Generating a coherent and logically structured narrative based on the associations between related news and user interests, thus engaging users in further reading of the news. Specifically, we propose Generative News Recommendation (GNR). First, we compose the multi-level representation of news and users by leveraging LLM to generate theme-level representations and combine them with semantic-level representations. Next, in order to generate a coherent narrative, we explore the news relationship and filter the related news according to the user preference. Finally, we propose a novel training method named UIFT to train the LLM to fuse the multiple related news in a coherent narrative. Extensive experiments show that GNR can improve the recommendation accuracy and eventually generate more personalized and factually consistent narratives.

## CCS CONCEPTS

• **Do Not Use This Code → Generate the Correct Terms for Your Paper**; *Generate the Correct Terms for Your Paper*; Generate the Correct Terms for Your Paper; Generate the Correct Terms for Your Paper.

## KEYWORDS

News Recommendation, Large Language Models, Generative Recommendation

**ACM Reference Format:**

Anonymous Author(s). 2018. Generative News Recommendation. In *Proceedings of Make sure to enter the correct conference title from your rights confirmation emai (Conference acronym 'XX)*. ACM, New York, NY, USA, 10 pages. https://doi.org/XXXXXXX.XXXXXXX

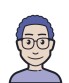

Clicked History

*1. Trump* attends Ivanka and Jared Kushner's wedding anniversary.

*2.* After push from *Perry*, backers got huge gas deal in *Ukraine*.

*3.* Should the *Department of Energy* and its federal jobs move to Kentucky? 2 senators think so.

(1) Traditional Recommender System: Top-K News List

*a)* *Rick Perry* said on Thursday that he plans to leave his post later this year after ... Perry's resignation comes amid scrutiny over his role in the *Trump* administration's dealings with *Ukraine*

*b)* Senate on Monday confirmed Dan Brouillette to be President Trump's second secretary of *Energy Department* ...

*c)* *Trump* announced Friday that he plans to nominate Dan Brouillette to replace *Rick Perry* ...

(2) **GNR**: Interest-aware Multi-news Narrative

Fuse News{*a*, *b*, *c*} : *Perry* announced that he will resign as the Secretary of *Department of Energy* later this year over *Ukraine* heats up, and *Trump* will nominate Dan Brouillette as the successor.

**Figure 1: Only recommending existing news in the news corpus is one of the limitations of traditional methods.**

## 1 INTRODUCTION

Online news platforms, such as Google News, have become crucial avenues for users to acquire daily information [5]. However, it is challenging for users to find interested content among a large number of articles. Hence, the news recommender system, which selects news based on user preference, is designed to improve the experience of user reading and alleviate the information overload problem [8].

Nevertheless, traditional news recommendation, encounter the following limitations: (1) News recommendation is a content-based task that mainly relies on semantic matching between candidate news and user preference. These methods only capture the explicit semantic relationship and ignore the implicit relationship which requires common-sense knowledge and reasoning ability to discover. For example, a news article about "Argentina's win over France was the greatest World Cup final ever" does not have any similar semantics with a news article about "Lionel Messi cements his place among the greats after winning epic duel against Kylian Mbappé". A user who is interested in the theme "Messi won the World Cup final" may have a high interest in reading these two news articles. However, finding the relationship between these two articles requires the knowledge of "Messi is a player of the Argentina national football team" and reasoning ability. (2) Existing

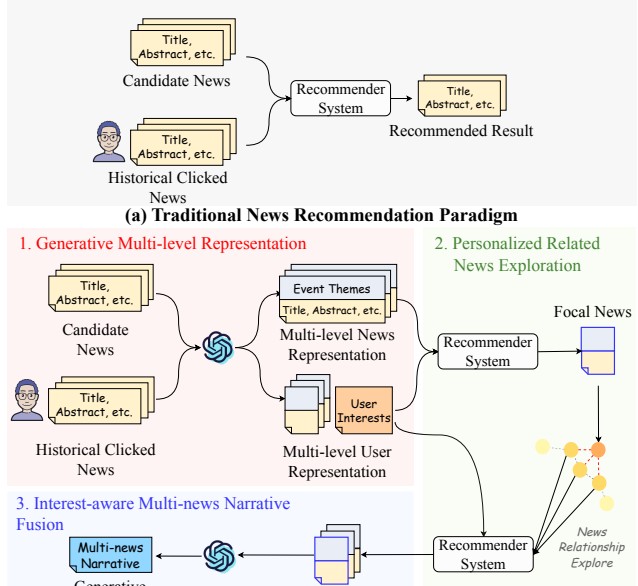

(a) Traditional News Recommendation Paradigm

1. Generative Multi-level Representation

2. Personalized Related News Exploration

3. Interest-aware Multi-news Narrative Fusion

(b) Generative News Recommendation Paradigm (Ours)

Figure 2: The differences between traditional news recommendation paradigm and our proposed generative news recommendation paradigm GNR.

news recommender systems can only recommend news from the news corpus in its original form. Users are required to read numerous lengthy news articles to gain an understanding of the overall context of events. Furthermore, users with different interests are presented with identical content without any personalization. Figure 1 shows an example of the recommended news list of existing methods. Although the recommended news list covers the main events of a user-preferred topic, users will read several long news articles with redundant information. A desired result form of news recommender system will be a concise paragraph that overviews the main events that the user interested in.

In this paper, we introduce a novel generative news recommendation paradigm (GNR). Our approach incorporates Large Language Model (LLM) as a generator to enhance news recommendation by precisely catering to user needs. As illustrated in Figure 2(a), traditional news recommendation methods perform semantic matching using candidate news and user representations, primarily composed of the user's history of clicked news. These methods subsequently present a list of news articles to the user in their original format. In contrast, our proposed paradigm employs LLM to generate multi-level representations for candidate news and user historical clicked news, as depicted in Figure 2(b). Then we explore the personalized related news set as the information source and finally generate a coherent and logically structured narrative to engage users in further reading of the news.

Specifically, the GNR consists of three modules. The first two modules aim at retrieving a news set that contains user-interested news and its related news, and the last module fuses the news in the set into a coherent narrative. (1) **Generative Multi-level**

**Representation**: Following the previous news recommendation methods [18, 20, 21], we first obtain semantic-level representations for both user and candidate news. Then we leverage the LLM to map news content and user profiles to theme-level representations. Finally, we combine these representations to multi-level representations. (2) **Personalized Related News Exploration**: In order to generate coherent narratives, we need to find a personalized and interrelated news set. There are three main steps in this module. We first conduct news ranking based on the user and news multi-level representations and obtain a focal news that matches user preference. Second, we propose to explore the logical relationships between news, aiming to discover more news related to the focal news. Since the second step introduces more related news that may not be interesting to the user, we conduct personalized filtering in the third step. Finally, we obtain a reference news set that encompasses both the main event context of focal news and takes into account the user preferences. (3) **Interest-aware Multi-news Narrative Fusion**: The primary objective of this module is to create a coherent and logically structured narrative that encapsulates the central theme of the reference news set. To enhance the alignment between the generated narrative and user interests, and to attract users to engage more with the content, we introduce the User Interest Alignment Fine-Tuning (UIFT) method, which adjusts the probabilities of multiple generated multi-news narratives by optimizing for ranking loss. Extensive experiments conducted on a benchmark dataset demonstrate that GNR improves recommendation accuracy and offers users more personalized multi-news narratives.

To sum up, our contribution can be summarized as follows:

• We propose a generative news recommendation paradigm (GNR), which introduces powerful LLM as the generator to make the recommended news meet user needs more precisely.

• In GNR, we design three modules to perform two sub-tasks: (I) Leveraging the internal knowledge and reasoning capability of LLM to retrieve a personalized related news set; (II) Generating a coherent and logically structured narrative, thus engaging users in further reading of the news.

• We propose a novel training method User Interest Alignment Fine-tuning (UIFT) which fine-tunes the LLM through ranking loss based on user interests.

• Extensive experiments on the MIND dataset demonstrate that the GNR can significantly improve the accuracy of recommendation systems and the generated narratives are more personalized to fulfill the user information needs.

## 2 RELATED WORK

### 2.1 Generative LLMs for Recommendation

Recently, LLMs have achieved great success in many natural language processing tasks due to their excellent natural language understanding and natural language generation capabilities [25, 26]. In addition, many studies have surfaced that LLMs can be used for recommendations due to their strong instruction following and common sense reasoning abilities [4, 11, 13].

Initially, Wang and Lim [14] proposed leveraging LLMs to solve the sequence recommendation task directly by prompting. It involves a three-step prompt, which includes user preference prompting, representative movie selection prompting, and recommendation prompting. Hou et al. [7] proposed using LLMs as a ranker, which converts the interaction history and candidate items in the recommendation task into natural language form, inputs them into LLMs together with the ranking instruction, and finds that LLMs have zero-shot ranking abilities. However, many researchers found that when LLMs are directly applied to recommendation tasks through prompting or in-context learning, there is a certain performance gap campared with existing recommendation models. The reason is that the pre-training tasks of LLMs are mainly related to NLP, and they lack training on recommendation tasks. In order to solve this problem, Bao et al. [3], Zhang et al. [24] proposed to improve the performance of LLMs for recommendation by instruction tuning, which involves generating natural language format instructions based on data from the recommendation task. Moreover, Bao et al. [2] proposed a two-step grounding framework (i.e., BIGRec) to assess LLMs' overall ranking capabilities instead of using restricted candidate sets.

In addition to directly using LLMs to conduct recommendation, some researchers propose to use generative LLMs to assist tradiotional recommender systems to recommend. Gao et al. [6] proposed Chat-REC leverage ChatGPT as the controller of the recommender system and the interface with the users. Chat-REC allows the users to express their needs proactively and also makes the recommendation more explainable. In addition, Chat-REC can also use the reasoning ability of LLMs to determine whether the system should recommend in the current round of dialog. Wang et al. [16] proposed a interactive evaluation approach based on LLMs, in order to solve the problem that the past evaluation metrics overemphasize the matching with the "ground-truth" items in conversational recommender system. Wang et al. [15] proposed a new generative recommender paradigm, which uses LLMs as the controller of the recommendation system to determine whether to recommend an existing item from the item corpus or to generate a new item through AI generator. Their method primarily uses LLMs as the controllers of the system, and utilizes the diffusion models to achieve the micro-video style transfer. Most of the previous works use LLMs as the recommender systems or the controllers. On the contrary, we leverage LLMs as the generator to provide mutli-level representations for better recommendation and generate the personalized and coherent multi-news narratives that assist users learn news events.

## 2.2 News Recommendation

The main task of the news recommendation is to recommend the news from the news corpus that is consistent with the user preferences, which can help users find the information they are interested in from the mass of news [20]. Compared with the ID-based recommendation tasks, the data in news recommendation contains a large amount of content information [23], so the development of news recommendation methods is also closely related to NLP. The core of the news recommender system includes the news encoder and the

user encoder. Wu et al. [17] mainly focused on enhancing news representation, and proposed a news recommendation approach with attentive multi-view learning (NAML), which uses word-level and view-level attention networks to select key information in news. An et al. [1] proposed a neural news recommendation approach with both long-term and short-term user representations instead of learning single-mode representations of users. Wu et al. [19] proposed a neural news recommendation approach called NRMS, which combines the multi-head self-attention mechanism in the news encoder and the user encoder. Following previous work, Wu et al. [21] used the more powerful pre-trained language models as backbone for the news encoder and user encoder. With the rise of LLMs, Liu et al. [9] proposed to utilize the leverage both open- and closed-source LLMs to enhance content-based recommendation, which includes the news recommendation.

However, these existing news recommendation methods are all based on human-written news corpus, which means they can only recommend raw news articles as-is. In our work, GNR can use LLMs to fuse multi-news narratives that more align with user interests.

## 3 TASK FORMULATION

The generative news recommendation can be formulated as retrieving a reference news set $\mathcal{N}^r$ from the news corpus $\mathcal{N}$ and generating a coherent narrative $n^m$ to overview the main events of these news. This reference news set must fulfill two key characteristics. Firstly, it should align with the user interests. Second, it should comprehensively mine related news by exploring implicit relationships among news, encompassing the full context of an event. And then the generated multi-news narrative $n^m$ can introduce the full context of news event. In this paper, we use $\mathcal{N} = [n_1, n_2, \ldots, n_k]$ to denote the whole news corpus which has $k$ news articles in total. The news recommendation system models the user preference based on the user's historical clicked news list $\mathcal{N}^h = [n_1^h, n_2^h, \ldots, n_i^h]$.

Then the recommendation system matches the user preference and news in the candidate list $\mathcal{N}^c = [n_1^c, n_2^c, \ldots, n_j^c]$ to predict the scores and outputs a focal news $n^f$ with the highest matching scores. Based on the focal news, we apply a filter to the whole news corpus $\mathcal{N}$ in order to find a reference news set $\mathcal{N}^r = [n^f, n_1^r, \ldots, n_{T-1}^r]$, which is both personalized and relevant. Then we fuse the reference news set $\mathcal{N}^r$ to obtain a multi-news narrative $n^m$ as the generative recommended result.

## 4 GNR METHOD

In order to help news recommendations better satisfy user needs, we propose Generative News Recommendation (GNR).

The GNR consists of three modules, as shown in Figure 2. **First**, we leverage the LLM to generate theme-level representations and combine with the semantic-level representations to obtain the multi-level representations, as shown in § 4.1. **Second**, we design a three-step pipeline to obtain a personalized and interrelated news set, in which we first rank the candidate news, then explore the relationship between news, and finally filter the related news set. Details are shown in § 4.2. **Third**, we fuse the news set to generate a brief multi-news narrative, which can assist user quickly learn about news events that interest him/her. Details are shown in § 4.3.

**Table 1: Example prompts for theme-level news representation generation**

| Instruction |
| --- |
| Based on the given news information, summarize what **topic(s)** the news is related to. Each news article is related to 1-3 topic, and each topic should not exceed five words. |
| **Input** |
| Title: "Trump says the Kurds 'are no angels' and the PKK are 'probably worse' than ISIS", Abstract: "President Trump defended his decision to withdraw U.S. forces from Syria, claiming that Turkey had been planning its invasion for some time and saying that the Kurds 'are no angels' ...... ", Category: "Politics" |
| **Output** |
| This news is related to **[Trump's decision on Syria]**, **[Kurds and PKK]**. |

**Table 2: Example prompts for theme-level user representation generation**

| Instruction |
| --- |
| You are asked to describe user interest based on his/her browsed news list. User interest includes the news **[categories]** and news **[topics]** (under each **[category]**) that users are interested in. |
| **Input** |
| News List: |
| {"ID": "News 1", "category": "sports", "topics": "Argentina football player Lionel Messi", "title": "Lionel Messi says he wants to ..."} |
| {"ID": "News 2", "category": "sports", "topics": "World Cup final", "title": "How the world reacted to 'the best World Cup final ever'"} |
| {"ID": "News 3", "category": "sports", "topics": "Lionel Messi won the World Cup", "title": "Lionel Messi cements his place among ..."} |
| {"ID": "News 4", "category": "sports", "topics": "World Cup final, Argentina team won the World Cup", "title": "Why Argentina's ..."} |
| **Output** |
| According to **[News 1, News 2, News 3, News 4]**, this user is interested in news about [sports], especially **[Lionel Messi, World Cup final, Argentina's victory in the World Cup]**. |

In this section, we introduce the detailed implementation process of the GNR.

## 4.1 Generative Multi-level Representation

News recommendation as a content-based task mainly focuses on performing recommendations through semantic matching. Existing news recommendation methods take the original content of the news as input to get the semantic-level representation of news and user using an encoder. However, these representations only stay at a low level and these methods ignore the higher-level association between news and users. Since a higher-level association between news requires domain knowledge and reasoning ability, we propose to use the LLM to generate a theme-level representation for both news and user profiles, and the theme-level representation aligns the news and user profiles at a higher-level. Finally, we combine the theme-level representation with the semantic-level representation to obtain a multi-level representation and promote more accurate matching.

*4.1.1 Theme-level News Representation.* Based on the in-context learning, we leverage the common-sense knowledge in the LLM to summarize the event themes for each news. For example, the original content of a news is "In the 2022 FIFA World Cup, Lionel Messi cements his place among the greats after winning epic duel against Kylian Mbappé". Then, the theme of this news is "Messi won the World Cup", which also serves as the theme-level representation of this news. Specifically, we first manually construct a prompt template, and then put the original news content as input, including news titles, abstracts and categories. The specific prompt construction is shown in Table 1.

*4.1.2 Theme-level User Representation.* To obtain the theme-level user representations, we reason the association within the historical clicked news list of each user using LLM and generate description about user profile containing several news themes. For example, if a user's historical clicked news list contains "Why Argentina's win over France was the greatest World Cup final ever" and "Lionel Messi cements his place among the greats after winning epic duel against Kylian Mbappé", we can infer that this user is interested in the event theme "Argentina won the World Cup". Additionally, for in-context learning, we manually construct a prompt template and input the multi-level news information, including news titles, abstracts, and themes. The specific prompt construction is shown in Table 2.

*4.1.3 Multi-level Representation Combination.* As mentioned above, we leverage the LLM to generate theme-level representations for news and user profiles. Meanwhile, we can get the semantic-level representation based on the original content of the news. Then we need to combine these two representations and provide the multi-level representations to the recommender system. Inspired by NAML [17], which is a widely known news recommendation method, we also use the multi-view attention network to fuse the representation:

$$\alpha_s = q_{v1}^T \tanh\left(\text{Linear}\left(e_s\right)\right), \alpha_t = q_{v2}^T \tanh\left(\text{Linear}\left(e_t\right)\right)$$
$$\alpha_s' = \frac{\exp\left(\alpha_s\right)}{\exp\left(\alpha_s\right) + \exp\left(\alpha_t\right)}, \alpha_t' = \frac{\exp\left(\alpha_t\right)}{\exp\left(\alpha_s\right) + \exp\left(\alpha_t\right)} \quad (1)$$
$$e_m = \alpha_s' e_s + \alpha_t' e_t,$$

where $q_{v_1}$ and $q_{v_2}$ both are attention query vector, $e_s$ is the embedding of semantic-level representation, $e_t$ is the embedding of

theme-level representation and $e_m$ is the embedding of multi-level representation. In the following, we default the user embedding $e^{user}$ and the news embedding $e^{news}$ are calculated based on the multi-level representations.

## 4.2 Personalized Related News Exploration

To ensure the coherence of the recommended narrative and cater to user interests, it is imperative to extract a personalized and interrelated news set, referred to as a "reference news set" from the news corpus $\mathcal{N}$. To achieve this goal, we have devised a three-step pipeline, which encompasses multi-level news ranking, news relation exploration, and personalized filtering. In the subsequent sections, we introduce the details of these three steps.

*4.2.1 Multi-level News Ranking.* To cater to user interests effectively, we adopt a candidate news ranking method similar to the conventional news recommendation paradigm. Instead of using the candidate news text and user historical clicked news text as input to the news recommender system, we utilize the multi-level user and news representation as input. To train the ranking model, we use the negative sampling and randomly select $K_{neg}$ non-clicked candidate news as negative samples. Then the probability of the user clicking the positive candidate news is:

$$\hat{y}_i = \text{softmax}\left(e_i^{user} \cdot e_i^{cand}\right),$$
$$p_i = \frac{\hat{y}_i^+}{\hat{y}_i^+ + \sum_{j=1}^{K_{neg}} \hat{y}_i^j}, \tag{2}$$

where $e_i^{user}$ is the embedding of multi-level user representations, $e_i^{cand}$ is the embedding of multi-level candidate news representations and $\hat{y}_i$ is the predicted ranking score of the candidate news. We optimize the probability of positive sample $p_i$ through log-likelihood loss:

$$\mathcal{L}_{ranking} = -\sum_{i \in S} \log\left(p_i\right), \tag{3}$$

where $S$ is the training dataset.

Then, we can obtain a ranking score for each candidate news, indicating the matching degree between the news semantic and user preference. However, the top news articles in the ranking list may not have any interrelation, and we cannot generate a coherent narrative based on these news articles. Thus, we only select the top-1 news as the focal news $n^f$ in the ranking list and use it to conduct the subsequent steps. Since this module is model-agnostic to the recommendation system, we have conducted extensive experiments using various recommendation methods.

*4.2.2 News Relation Exploration.* After obtaining the focal news $n^f$ that is most relevant to the user's interests, we need to find some related news from the news corpus to form a recommended narrative. In order to ensure the coherence of the narrative, we need to find some interrelated news. Therefore, we implement a news relationship classifier to judge whether two news articles are related and set a relationship threshold $\alpha$. In order to train the model, we collected a set of related news data pairs from news websites to form a training dataset. We construct a news relation classifier based on Siamese networks [12] and train the model using contrastive learning loss. The positive pair is a pair of news that is

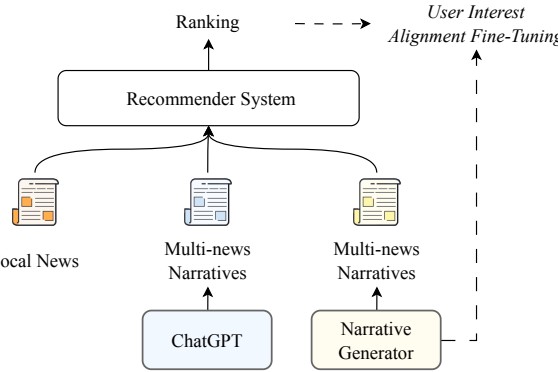

**Figure 3: The framework of UIFT method.**

related to each other, and we randomly select five unrelated news for the news to construct negative data pairs. The loss function is formulated as follows:

$$\mathcal{L}_{classify} = \max\left(\left\|e_i^{news} - e_j^{news}\right\| - \left\|e_i^{news} - e_k^{news}\right\| + \epsilon, 0\right), \tag{4}$$

where $e_j^{news}$ is a positive news (*a.k.a.,* related news), and $e_k^{news}$ is a negative news (*a.k.a.,* unrelated news). After model training, we use this news relation classifier to explore a set of related news $\mathcal{N}^{rel} = \{n_1^{rel}, n_2^{rel}, \dots, n_j^{rel}\}$ related to the focal news $n^f$ from the news corpus.

*4.2.3 Personalized Filtering.* Since the related news set $\mathcal{N}^{rel}$ is explored based on the theme and semantic relationship between news, which ignores the user preference. In this module, we propose to personalized filter the related news set based on the traditional recommender systems. We use the related news set $\mathcal{N}^{rel}$ as the candidate set and compute the matching score $\hat{y}_i$ between related news embedding $e_i^{rel}$ and user embedding $e_i^{user}$ as follows:

$$\hat{y}_i = \text{softmax}\left(e_i^{user} \cdot e_i^{rel}\right). \tag{5}$$

Then we select $T-1$ news with the highest matching score. And together with the focal news $n^f$ obtained above, we form the reference news set $\mathcal{N}^r$.

## 4.3 Interest-aware Multi-news Narrative Fusion

Conventional news recommendation methods typically recommend news articles from the corpus in their original form. Consequently, when users express interest in a specific news event, they must peruse a multitude of related news articles to gain an understanding of the pertinent content. This process is notably time-consuming and results in a suboptimal user experience. Simultaneously, personalization diminishes as recommender systems deliver the same news content to users with varying interests. To address this challenge, we propose the fusion of multi-news narratives based on user interests. In this section, we first introduce the process of multi-news narrative fusion and subsequently introduce the UIFT method, designed to tailor the generated narrative more closely to user interests.

First, to ensure the coherence and readability of the narrative, we use the focal news article $n^f$ (as introduced in § 4.2.2) as the central point of the multi-news narrative. Then the goal of the generated narrative is to extract the key facts of the news set $\mathcal{N}^\nabla$ that align with user interests and fuse these key facts around the focal news $n^f$. An illustrative example of this fusion process can be found in § A.1.

Despite the robust in-context learning capabilities of general black-box LLM, they still exhibit deficiencies in capturing user interests, a crucial aspect of the recommendation task. We first employ ChatGPT to generate the multi-news narrative and conduct supervised fine-tuning of our multi-news narrative generator to distill the ability and knowledge from ChatGPT to our narrative generator. As shown in Figure3, we propose User Interest Alignment Fine-Tuning (referred to as UIFT), aimed at enhancing the personalization of LLM-generated multi-news narratives. UIFT trains the LLM by incorporating ranking loss, thereby aligning its ranking for multiple news narratives with user interests. To train this model, we collect a training pair containing a multi-news narrative generated by Chat-GPT $n_i^{ChatGPT}$, a multi-news narrative fused by our multi-news narrative $n_i^{GNR}$, and the focal news $n_i^f$. Next, UIFT first allows the our narrative generator $\pi$ trained with supervised fine-tuning to predict the conditional probability $p_i$ for each news:

$$p_i = \frac{\sum_t \log P_\Pi\left(s_{i,t} \mid n, s_{i<t}\right)}{\|s_i\|}, \tag{6}$$

where $n$ is the news in the set $\{n_i^{ChatGPT}, n_i^{GNR}, n_i^f\}$.

In UIFT, the model learns to give higher probabilities to more personalized news, thus aligning with user interests. Specifically, we employ a well-trained news recommender system to rank the news set $\{n_i^{ChatGPT}, n_i^{GNR}, n_i^f\}$. Based on the predicted score computed by the recommender system, we can get an ordering between the news (i.e., $r_i^{GNR} < r_i^{ChatGPT} < r_i^f$) and the $r_i \in \{1, 2, 3\}$ denotes the ranking of the news. When $r_i$ is smaller, it means that the corresponding news is more aligned with the user interests.

Our training goal for the UIFT is to give the larger probability $p_i$ to news with better ranking $r_i$. We achieve this through ranking loss:

$$L_{\text{rank}} = \sum_{r_i < r_j} \max\left(0, p_i - p_j\right), \tag{7}$$

## 5 EXPERIMENT

In this section, we conduct extensive experiments to answer the following research questions:

**RQ1**: How much accuracy improvement can GNR bring to the recommendation models by combining multi-level representations?

**RQ2:** Can GNR generate personalized and factually consistent multi-news narratives?

**RQ3**: When exploring the relationship between news, how does the relationship threshold $\alpha$ affect the performance?

**RQ4**: When retrieving the reference news set $\mathcal{N}^r$, how does the number of reference news $T$ affect the performance?

### 5.1 Datasets

We conduct experiments on the MIND dataset [22], which is a large-scale dataset for news recommendation. MIND dataset contains the news dataset and behaviors dataset. Each news in MIND contains news title, abstract and etc. Each behavior consist of historical clicked news list $\mathcal{N}^h$ and candidate news list $\mathcal{N}^c$. For the news dataset, we first filtered 5145 news articles from MIND-Large under "politics" category, which is more appropriate for the scenario of GNR. We will extend the GNR to experiment under more news categories in the future. For the behaviors dataset, we first filtered behaviors to ensure all news in historical clicked list and candidate list belong to the politics news dataset. When fusing multi-news narratives, we are constrained by the maximum input length of LLM. Therefore, we filter behaviors to ensure that the length of the past clicked news list falls within the range: $5 < |\mathcal{H}| < 15$. We then structured two separate datasets for training the recommender system and the multi-news narrative generator respectively. The sizes of the training, validation, and test sets for the recommender system are 43,232, 4,800, and 6,713. Similarly, for the multi-news narrative generator, the sizes of the training, validation, and test sets are 43,232, 4,800, and 6,713.

Furthermore, to train the classifier for news relationship exploration, we employed a web crawler to extract news articles from the CNN website[1]. We also scraped the relevant news from each news page to construct the dataset, considering these as related news. While retrieving the reference news, we used them to augment the related news and enrich our news database.

### 5.2 Evaluation Metrics

The GNR proposed two sub-tasks: (1)Retrieving personalized related news sets; (2)Fusing coherent multi-news narratives. We evaluate the performance of these two sub-tasks separately.

For the first sub-task, we evaluate whether the theme-level representations generated by GNR can promote recommendation accuracy. So we leverage AUC(Area Under the ROC Curve), MRR(Mean Reciprocal Rank), NDCG@K(Normalized Discounted Cumulative Gain) where K=5 as the evaluation metrics.

For the second sub-task, because GNR is a novel paradigm, there is no previous benchmark. Therefore, we propose two automatic metrics to evaluate the personalization and consistency of multi-news narratives: (1)**Winning Rate** that evaluate whether the fused multi-news narratives are more personalized than the corresponding focal news. We first calculate the predicted scores of the multi-news narrative and the focal news based on a well-trained news recommendation model. And the inputs of the recommender system are the multi-level news representations and user representations. We mark a situation as a "Win" when the predicted score of a multi-news narrative surpasses that of the focal news, interpreting this as an indication that the multi-news narrative aligns better with the user's interests. Then we compute the winning rate across the entire test dataset. (2)**Consistency Rate**, which is calculated between the reference news sets and the multi-news narratives. During evaluation, we feed both a reference news set and a multi-news narrative into the evaluator. The evaluator then determines whether the reference news set and the multi-news narrative are

---

[1]https://edition.cnn.com/

**Table 3: Performance of recommendation accuracy. We experiment with different combinations of news representations and user representations. "Sem" denotes semantic-level representation. "Multi" denotes multi-level representation.**

| Input Type | | NRMS | | | PLM4NR (title) | | | PLM4NR (title and abstract) | | |
|---|---|---|---|---|---|---|---|---|---|---|
| News | User | NDCG@5 | AUC | MRR | NDCG@5 | AUC | MRR | NDCG@5 | AUC | MRR |
| Sem | Sem | 61.98 | 69.22 | 54.44 | 62.36 | 69.91 | 54.44 | 61.11 | 68.36 | 53.70 |
| Sem | Multi | 63.02 | _71.16_ | 55.42 | 62.98 | **70.38** | 55.21 | 61.78 | 68.77 | 54.35 |
| Multi | Sem | _63.14_ | 69.72 | _55.99_ | _63.02_ | 69.95 | _55.61_ | _62.76_ | _69.26_ | _55.52_ |
| Multi | Multi | **63.57** | **71.34** | **56.10** | **63.46** | _70.31_ | **56.03** | **62.99** | **69.42** | **55.72** |

**Table 4: Performance of multi-news narrative Fusion. Win Rate is evaluated by well-trained PLM4NR.**

| Generator | Win Rate | Consistency |
|---|---|---|
| ChatGPT | 72.80 | 96.63 |
| Ours(SFT) | 65.13 | 96.52 |
| Ours(UIFT) | 74.54 | 96.57 |

consistent. Subsequently, we compute the consistency rate across the entire test dataset. And inspired by Luo et al. [10], we use the ChatGPT(gpt-3.5-turbo) as the evaluator to determine consistency.

## 5.3 Baseline Models

We evaluate GNR against following news recommendation methods: **(1) NRMS[18]** leverages the multi-head self-attentions to learn news representations and capture the relatedness between the news; **(2) PLM4NR (title)[21]** uses pre-trained language models to model news representations from news title; **(3) PLM4NR (title and abstract)[21]** is similar to PLM4NR(title), but it learns news representations from news titles and news abstracts not just titles. Besides, all of the encoders in the above baseline models are DistilBERT in the experiments.

## 5.4 Implementation Details

For theme-level representations generation, we select the ChatGPT(gpt-3.5-turbo) as the backbone model. For news relationship exploration, we use the SBERT [12] as the backbone model. During the training, we use AdamW as the optimizer and the learning rate is set to 1e-5. When clustering, we set hyperparameter $\alpha = 0.8$. For multi-news narratives fusion, we use the LLaMA 7B as the backbone model. Meanwhile, we treat the best recommended news (i.e., ground truth) as the focal news in the reference news set to better compare whether the multi-news narratives fused by GNR can better match user preferences and set the number of reference news $T$ cannot be larger than 4. During the training of traditional news recommender systems, the learning rate is set to 5e-5. During the supervised fine-tuning of LLaMA, the learning rate is set to 5e-5. During the UIFT, the learning rate is set to 1e-5. The maximum length of input and output are 1280 tokens and 256 tokens separately. Besides, all LLaMA-based experiments are conducted 8 80GB Nvidia A800 GPUs, and others on a 2 24G Nvidia 3090 GPUs.

## 5.5 Performance of Recommendation Accuracy(RQ1)

We first evaluate the recommendation accuracy when using the multi-level representations as input, and report the results in 3. From the results, we can have the following observations.

*5.5.1 Result Analysis.* Compared to the accuracy of using the semantic-level representation only, the performance of the backbone models increase significantly when using the multi-level news and user representations. For example, the NRMS model improved from 61.98 to 63.57 on metric NDCG@5 and improved from 69.22 to 71.34 on metric AUC. This suggests that with the help of common-sense knowledge in LLM, the news recommender systems can capture more levels of relationships between the news and user interests. These relationships can help systems better match the candidate news and users, and improves the recommendation accuracy.

Besides, the multi-level representations are more effective for the backbone models that use only news titles as input than the backbone models that use both news titles and abstracts as input. The reason is that we used news titles and abstracts to generate theme-level representations for user and news, so the theme-level representations contain more news information. And the theme-level representations are very concise so that the additional information does not introduce noise.

*5.5.2 Ablation Study.* To separately evaluate the effect of theme-level news representation and theme-level user representation, we conducted ablation experiments with two settings: (1)using multi-level news representation and semantic-level user representation; (2)using semantic-level user representation and multi-level news representation. As observed, the recommendation models outperform the baseline models when utilizing theme-level news (or user) representations. However, their performance is not as good as the models that employ both theme-level news representations and user representations. So these results illustrate that the theme-level representations for both news and users can promote the performance of the recommendation models.

## 5.6 Performance of multi-news narrative fusion(RQ2)

In this part, we evaluate whether the multi-news narratives fused by GNR can perform better than the focal news and whether the multi-news narratives are factually consistent with the corresponding reference news. The results report in Table 4.

When using ChatGPT as the generator, GNR has great performance in both Winning Rate and Consistency Rate. It illustrates that ChatGPT is able to fuse the coherent and personalized multinews narrative. We consider this is due to ChatGPT's excellent in-context learning capability, allowing it to perform our desired task without additional fine-tuning.

When our narrative generator is trained only through supervised fine-tuning, it can effectively fuse news and generate coherent narratives. Nevertheless, its performance in Winning Rate is considerably less effective than that of ChatGPT. However, when our narrative generator is trained through UIFT, the resulting fused multi-news narratives can achieve superior personalization while maintaining good consistency. We attribute this to the fact that in UIFT, we align the probabilities of our narrative generator with user interests via ranking loss. This alignment aids the model in better understanding user interests during the narrative fusion process.

**Table 5: The impact of relationship threshold $\alpha$ on multi-news narratives**

| relationship threshold $\alpha$ | Consistency Rate |
| --- | --- |
| 0.6<$\alpha$<0.7 | 0.66 |
| 0.7<$\alpha$<0.8 | 0.85 |
| 0.8<$\alpha$ | 0.98 |

## 5.7 Threshold of News Relationship(RQ3)

As described in § 4.2, we design to retrieve a reference news set which needs to be personalized and interrelated. Therefore, we plan to explore the news relationship and find a news set related to the focal news. In this process, we define a relationship threshold $\alpha$ for determining if two news articles are related. We think this threshold $\alpha$ would has an effect on the quality of multi-news narrative which is final fused based on the reference news set. Therefore, in order to evaluate the impact of this threshold on the fused narrative, we set different threshold ranges to conduct the experiments separately. 0.6<$\alpha$<0.7 means that the related news set must contain several news with similarity in [0.6, 0.7], and 0.7<$\alpha$<0.8 means that the related news set must contain news with similarity in [0.7, 0.8], and $\alpha$>0.8 means that the related news set must only contains news with similarity greater than 0.8. Among all compliant samples, we randomly selected 100 samples for each interval separately. However, there are only 82 samples that comply with the condition 0.6<$\alpha$<0.7, so we select all of them. Finally, we evaluate the Consistency Rate of the narratives under each setting, and the results are shown in Table5.

From the results, we can see that the relevance of the reference news set has an impact on the consistency of the fused narrative. We hypothesize that this is due to the fact that when the relevance of the reference news set is low, the generator is unable to reason correctly about the associations between the reference news, leading to the hallucination generation.

## 5.8 The Number of Reference News(RQ4)

In this part, we hypothesize that the number of reference news $T$ determines the quality of input information, thus affecting the

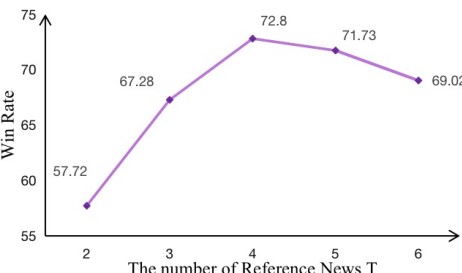

**Figure 4: The impact of the number of Reference News $T$ on Multi-News Narratives.**

final generation quality. Therefore, we experiment how the number $T$ promote or reduce the performance of fusion. Considering the limitation of the LLM input length, we set $T$=2,3,4,5,6 respectively to conduct experiments and evaluate the fused narratives in each setting.

As we can see in Figure 4, with the increasing number of reference news, the Winning Rate increase first and then decrease, and reach the optimum at $T$=4. This observation demonstrates when $T$ is less than 4, the reference news set contains insufficient information to adequately cover the main ideas of the events and cater to user interests. When $T$ is more than 4, there is too much information in the reference news set that may not be interesting to the user. Therefore, it introduces noise during the fusion, resulting in the fused narratives failing to meet user need.

## 6 CONCLUSION

In this paper, we introduce a novel generative news recommendation paradigm GNR, aimed at enhancing news recommendation and fulfilling user needs more precisely using LLM. By harnessing the internal knowledge and reasoning capabilities of LLM, we generate theme-level representations for news and users. These representations are then combined with semantic-level representations to create multi-level representations, which significantly improve recommendation performance. Subsequently, we explore the relationships between news articles to find a personalized related news set based on multi-level representations. Afterward, we fuse the personalized related news using LLM to create coherent and logically structured multi-news narratives, engaging users further in their news consumption. Extensive experiments conducted on MIND datasets demonstrate that our GNR enhances recommendation performance and generates personalized and factually consistent narratives.

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

# A  APPENDIX

## A.1  Example prompts for multi-news narrative generation

Received 20 February 2007; revised 12 March 2009; accepted 5 June 2009

| Instruction |
|---|
| You are a personalized text generator. First, I will provide you with a news list that includes both the **[main news]** and **[topic-related news]**. Second, I will provide you with user interests, including the **[categories]** and **[topics]** of news that the user is interested in. Based on the input news list and user interests, you are required to generate a **{personalized news summary}** centered around the **[main news]**. |

| Input |
|---|
| News List: |
| {"ID": "Main News", "title": "Lionel Messi says he wants to continue 'living a few more games being world champion'", "abstract": "Ever since he was a young boy growing up in Rosario, Lionel Messi's ultimate dream ...", "topic": "Messi ..."} |
| {"ID": "Topic-related News 1", "title": "How the world reacted to 'the best World Cup final ever'", "abstract": "This World Cup final was a game that seemed to defy comprehension, conventions and any attempt ...", "topic": "the World Cup final..."} |
| {"ID": "Topic-related News 2", "title": "Lionel Messi cements his place among the greats after winning epic duel against Kylian Mbappé", "abstract": "Now there can be no arguments. Now there should be no debate. In the ...", "topics": "Lionel Messi ..."} |
| {"ID": "Topic-related News 3", "title": "Why Argentina's win over France was the greatest World Cup final ever", "abstract": "It seems only yesterday that Enner Valencia was swatting aside Qatar in the 2022 World Cup...", "topics": "World Cup ..."} |
| User Interest: |
| This user is interested in news about[sports], especially [Lionel Messi, Argentina national football team]. |

| Output |
|---|
| "title": "Lionel Messi Desires to Continue 'Living a Few More Games Being World Champion", "category": "sports", "topic": "Lionel Messi, Argentina national football team, World Cup final", |
| "abstract": "**Lionel Messi**, who recently achieved his lifelong dream of winning the **World Cup for Argentina**, expressed his desire to continue 'living a few more games being world champion.' This victory, marked by several brilliant individual performances from **Messi**, has been celebrated globally. The final match, which included an intense duel with Kylian Mbappé, is being hailed as one of the most memorable **World Cup finals**." |

