# OpenReview forum: "Generative News Recommendation"
_ACM.org/TheWebConf/2024/Conference — TheWebConf24 Oral_

### Official Review · Reviewer_TUCw · 2023-11-05

**Novelty:** 5
**Technical Quality:** 4

**Review:**

This paper introduces the concept of "Generative News Recommendation", which aims to generate coherent narratives for users. It consists of a three-stage pipeline: 1. news and user summarization; 2. news exploration; and 3. narrative fusion.

The entire article's writing is passable at best, with some parts requiring multiple readings to comprehend.

Pros:
- Probably a new field to explore, enhancing user reading experience.
- A nice incorporation with LLMs by using its powerful comprehension and summarization ability.
- A good performance displayed in the experiments.

Cons:
- I doubt the effectiveness of the news exploration and narrative fusion. In the reality, users tend to read news of multiple categories in his/her browsing session and care multiple events all at once. Therefore, it would be hard to select the **focal news** and thus hard to perform news exploration.
- Missing an important baseline [1] you mentioned in the related work, which also produces item-level (content summarizer) and user-level (user profiler) representations in news recommendation domain. You can also include other research that combining LLMs with news recommendation [2][3].
- If I understand correctly, the first row of the experimental results in Table 3 (Sem News and Sem User) did not use data generated by ChatGPT. I am doubleful of the efficacy of the experiment involving PLM4NR variant that includes abstract, as it may not outperform the title-only version.
- Clarity: Line 300. PLM used in [4] is only used as the backbone of the news encoder.

[1] ONCE: Boosting Content-based Recommendation with Both Open- and Closed-source Large Language Models
[2] LKPNR: LLM and KG for Personalized News Recommendation Framework
[3] A Preliminary Study of ChatGPT on News Recommendation: Personalization, Provider Fairness, Fake News
[4] Empowering News Recommendation with Pre-trained Language Models

**Questions:**

Q1: The authors only select a small-scale subset (i.e., politics) of the MIND dataset. Is it reasonable to provide recommendations for a single category of news? Please first explain. If the answer is no, consider using a larger dataset and providing users with solutions that cater to multiple interests. Otherwise, please conduct more experiments on multiple subsets (e.g., travel, lifestyle, crime).

Q2: Can you compare GNR with the GENRE framework proposed by ONCE[1], which also use ChatGPT to generate news summary and user profile? And what are the differences?

Q3: Please refer to the third Cons and anaylse the experiments of the PLM4NR variants?

**Ethics Review Description:**

null

**Reviewer Confidence:**

4: The reviewer is certain that the evaluation is correct and very familiar with the relevant literature

**Scope:**

3: The work is somewhat relevant to the Web and to the track, and is of narrow interest to a sub-community

---

### Official Review · Reviewer_Fi73 · 2023-11-23

**Novelty:** 6
**Technical Quality:** 5

**Review:**

The paper proposes a new approach to news recommendation called Generative News Recommendation (GNR), which uses internal knowledge and reasoning capabilities of a Large Language Model to match candidate news with user representation, and then generates a coherent narrative based on the associations between related news and user interests. The approach, called UIFT, is shown to improve recommendation accuracy and result in more personalized and factually consistent narratives.

The idea is novel, which not only performs personalized matching based on LLM but also generate narratives based on related news set. It opens new research opportunites that combining recommendation and generation with the power of LLMs. However, it will be better to make more comprehensive experiments. For example, I am interested in the comparison between personalized narratives generation and non-personalized news summarization/generation based on LLM. I recommend the authors to provide more generated examples for readers.

**Questions:**

How much difference does the personalization make in narratives generation?

**Reviewer Confidence:**

4: The reviewer is certain that the evaluation is correct and very familiar with the relevant literature

**Scope:**

4: The work is relevant to the Web and to the track, and is of broad interest to the community

---

### Official Review · Reviewer_cMUN · 2023-11-27

**Novelty:** 5
**Technical Quality:** 5

**Review:**

This paper introduces a novel generative news recommendation paradigm GNR, aimed at enhancing news recommendation and fulfilling user needs more precisely using LLM. Extensive experiments conducted on MIND datasets demonstrate that the GNR enhances recommendation performance and generates personalized and factually consistent narratives.
Strengths:
1.	Overall, the proposed approach is well designed and the results quite convincing.
2.	This paper is well written and organized.
3.	The experiments are very solid.
Weakness:
1.	The dimensions of all vectors and matrices should be explicitly provided. Additionally, for clarity, vectors should be represented using bold lowercase letters, and matrices should be denoted by bold uppercase letters.
2.	The description of 'Multi-level Representation Combination' might be imprecise as the paper only mentions two levels. Typically, 'multi-level' implies three or more levels. Therefore, the terminology used in discussing the representation combination should accurately reflect the number of levels addressed in this context.
3.	Some operations are unclear and inaccurate, such as,
a)	In Eq. (1), there is no clarification for 'Linear,' and it is unclear whether \alpha is a vector or a scalar weight;
b)	In Eq. (2), it is imperative to ensure that the computation of $p_i$ does not encounter a division by zero scenario. For instance, incorporating a regularization term in the denominator could address this concern;
c)	In Eq. (3), the base of the logarithmic function should be explicitly stated, specifying whether it is a logarithm with base 2 or the natural logarithm with base e. This clarification should also be applied to Eq. (6);
d)	In Eq. (4), the term $e_j^{\text{news}}$ denotes the positive new embedding rather than the positive news itself, and the interpretation of $\varepsilon$ requires further elaboration;

**Questions:**

NA

**Reviewer Confidence:**

4: The reviewer is certain that the evaluation is correct and very familiar with the relevant literature

**Scope:**

4: The work is relevant to the Web and to the track, and is of broad interest to the community

---

### Official Review · Reviewer_7iL9 · 2023-11-28

**Novelty:** 5
**Technical Quality:** 5

**Review:**

This paper presents Generative News Recommendation. First, it utilizes a Large Language Model (LLM) for high-level matching between candidate news and user representation. This involves generating multi-level representations of news and users. Second, it generates a coherent narrative by exploring news relationships, filtering related news based on user preferences, and using a novel training method called User Interest-guided Fusion Training (UIFT) to train the LLM to merge multiple related news into a coherent story.

Pros:

1. It is interesting to incorporate LLMs into news recommendation.

2. The experimental results show some improvement.

Cons:

1. The paper writing quality should be improved. For example, it is difficult to understand Figure 2.

2. Only one datasdet is used for experiments, which is not sufficient.

3. Only two baseline methods are compared. More baselines should be compared.

4. The motivation of this work, illustrated in introduction, seems to be well handled by knowledge graph incorporated news recommendation methods. Why not compare with KG-based news recommendation methods?

**Questions:**

Can knowledge graph incorporated news recommendation methods handle the problem illustrated in the introduction section? Can you make some comparison with them?

**Reviewer Confidence:**

2: The reviewer is willing to defend the evaluation, but it is likely that the reviewer did not understand parts of the paper

**Scope:**

3: The work is somewhat relevant to the Web and to the track, and is of narrow interest to a sub-community

---

### Decision · Program_Chairs · 2024-01-22

**Decision:**

Accept (Oral)

**Comment:**

This work presents a novel paradigm to make news recommendations with generative models. The paradigm was systematically designed and clearly presented. Experimental results are capable of demonstrating the effectiveness of the proposed paradigm. Reviewers offer valuable suggestions and the authors' response provide new data that should be incorporated into this paper.